# Protective Effect of Memantine on Bergmann Glia and Purkinje Cells Morphology in Optogenetic Model of Neurodegeneration in Mice

**DOI:** 10.3390/ijms22157822

**Published:** 2021-07-22

**Authors:** Anton N. Shuvaev, Olga S. Belozor, Oleg I. Mozhei, Elena D. Khilazheva, Andrey N. Shuvaev, Yana V. Fritsler, S. Kasparov

**Affiliations:** 1Institute of Molecular Medicine and Pathobiochemistry, Voyno-Yasenetsky Krasnoyarsk State Medical University, 660022 Krasnoyarsk, Russia; olsbelor@gmail.com (O.S.B.); elena.hilazheva@mail.ru (E.D.K.); 2Institute of Living Systems, Immanuel Kant Baltic Federal University, 236041 Kaliningrad, Russia; vulpecula999@gmail.com; 3Siberian Federal University, 660041 Krasnoyarsk, Russia; andrey.n.shuvaev@gmail.com (A.N.S.); fri.yana@mail.ru (Y.V.F.); 4School of Physiology, Pharmacology and Neuroscience, University of Bristol, Bristol BS8 1TD, UK; Sergey.Kasparov@Bristol.ac.uk

**Keywords:** astrogliosis, optogenetics, Bergmann glia, ataxia, cerebellar neurodegeneration, glutamate uptake, excitotoxicity

## Abstract

Spinocerebellar ataxias are a family of fatal inherited diseases affecting the brain. Although specific mutated proteins are different, they may have a common pathogenetic mechanism, such as insufficient glutamate clearance. This function fails in reactive glia, leading to excitotoxicity and overactivation of NMDA receptors. Therefore, NMDA receptor blockers could be considered for the management of excitotoxicity. One such drug, memantine, currently used for the treatment of Alzheimer’s disease, could potentially be used for the treatment of other forms of neurodegeneration, for example, spinocerebellar ataxias (SCA). We previously demonstrated close parallels between optogenetically induced cerebellar degeneration and SCA1. Here we induced reactive transformation of cerebellar Bergmann glia (BG) using this novel optogenetic approach and tested whether memantine could counteract changes in BG and Purkinje cell (PC) morphology and expression of the main glial glutamate transporter—excitatory amino acid transporter 1 (EAAT1). Reactive BG induced by chronic optogenetic stimulation presented increased GFAP immunoreactivity, increased thickness and decreased length of its processes. Oral memantine (~90 mg/kg/day for 4 days) prevented thickening of the processes (1.57 to 1.81 vs. 1.62 μm) and strongly antagonized light-induced reduction in their average length (186.0 to 150.8 vs. 171.9 μm). Memantine also prevented the loss of the key glial glutamate transporter EAAT1 on BG. Finally, memantine reduced the loss of PC (4.2 ± 0.2 to 3.2 ± 0.2 vs. 4.1 ± 0.3 cells per 100 μm of the PC layer). These results identify memantine as potential neuroprotective therapeutics for cerebellar ataxias.

## 1. Introduction

Bergmann Glia (BG) are highly specialized astrocytes of the cerebellar cortex. Astrocytes, including BG, play a key role in glutamate uptake through excitatory amino acid transporter type 1 (EAAT1) [1]. Neurodegenerative diseases, such as spinocerebellar ataxias (SCA), evoke pathological transformation of BG, manifested by increased thickness and decreased length of its processes. This reactive phenotype also compromises glutamate uptake from synapses between parallel fibers (PF) and Purkinje cells (PC) [2,3]. As a result, glutamate hyperactivates synaptic and extrasynaptic NMDA receptors, leading to excessive Ca^2+^ entry and triggers PCs apoptosis and death [4], as demonstrated previously by our studies [5] and other evidence [6]. At present, there is no treatment for SCA. Memantine is a low-affinity, voltage-dependent noncompetitive antagonist of NMDA receptors approved for use in Alzheimer’s disease [7]. This mechanism could potentially also be suitable for the treatment of SCA. This, however, requires additional evidence, specifically focusing on cerebellar neurodegeneration [8]. We have recently developed and published a model of cerebellar neurodegeneration based on chronic optogenetic activation of BG with the light-sensitive cationic channel channelrhodopsin- 2 (ChR2) [5]. Chronic photostimulation of ChR2 leads to reactive changes in BG and excitotoxicity against PCs. This excitotoxicity was associated with reduced EAAT1 activity in BG [5]. The present study was undertaken to test whether memantine can reduce neurodegenerative effects caused by chronic optogenetic activation of BG.

## 2. Results

### 2.1. Memantine Prevents Pathological Changes in BG Morphology after Chronic Optogenetic Activation

We used the previously described model of cerebellar astrogliosis and neurodegeneration ([5] and Figure 1). After 4 days of photostimulation, anti-GFAP (astrocytic marker) staining/expression was strongly increased while anti-Calbindin (PCs marker) staining/expression was greatly reduced compared to the animals, which were not photostimulated (Figure 2A,B). Memantine (~90 mg/kg/day) strongly decreased the build-up of the anti-GFAP staining/expression (Figure 2C).

After 4 days of photostimulation, the thickness of BG processes expressing ChR2 increased from 1.57 ± 0.05 μm to 1.81 ± 0.05 μm, *p* = 0.003 and their length was decreased from 186.0 ± 8.1 μm to 150.8 ± 3.6 μm, *p* = 0.002 (Figure 3A,B,D–F). Memantine prevented BG reactivity, reducing the thickness of BG processes (1.62 ± 0.04 μm) *p* = 0.003 and restoring their length (171.9 ± 5.4 μm), *p* = 0.005 (one-way ANOVA followed by Tukey’s post hoc test; Figure 3A,B,D–F).

### 2.2. Memantine Rescued PC Morphology and Synaptic Transmission in PF-PC Synapses after Chronic Photostimulation of BG

After 4 days photostimulation, PCs number decreased from 4.2 ± 0.2 to 3.2 ± 0.2 per 100 μM of Purkinje Cell Layer (PCL) length (*p* = 0.011; Figure 3C,G). The ML thickness was reduced in these animals from 165.4 ± 6.9 μm to 139.2 ± 5.0 μm (*p* = 0.0015; Figure 3D,H). Memantine prevented PC number decline (4.1 ± 0.3 cells per 100 μm of PCL length; *p* = 0.009) and ML thickness loss (171.9 ± 5.4; *p* = 0.043 compared to untreated mice); (Figure 3C,D,G,H). 

Using patch clamp, we estimated the capacitance of PCs which is directly related to the size of their membrane. The average capacitance of PCs in mice without photostimulation was 725.1 ± 39.5 pF (*n* = 8 cells from 3 mice) and light stimulation reduced it to 494.2 ± 63.2 pF (*n* = 10 cells from 3 mice; *p* = 0.007; Table 1). Memantine significantly prevented PC capacitance loss (682.2 ± 24.5 pF, 7 cells from 3 mice, *p* = 0.38 compared to animals without stimulation; Table 1). These data demonstrate that memantine largely prevents pathological changes in the cerebellar cortex caused by chronic specific optogenetic stimulation of BG. 

We tested whether chronic photostimulation affects synaptic transmission in PF PCs synapses. In these experiments the amplitude PF EPSC’s dropped from 298.1 ± 64.6 (*n* = 8 cells from 3 mice) to 141.0 ± 23.1 (*n* = 10 cells from 3 mice; *p* = 0.033; Table 1). In memantine-treated mice, PF EPSC amplitude was 195.0 ± 33.7, *n* = 7 cells from 3 mice, and this was not significantly different from unstimulated animals (*p* = 0.19). To investigate the glutamate uptake, we recorded PF EPSC after a train of 10 stimuli delivered to PF. PF EPSC’s amplitudes were not significantly different between the groups (Figure 4A). However, the chronic photostimulation prolonged the decay time of PF EPSC (τ EPSC) from 16.0 ± 2.0 (*n* = 12 cells from 3 mice) to 39.1 ± 5.1 (*n* = 8 cells from 3 mice; *p* = 0.00014). This was prevented by memantine (22.2 ± 3.4 pF, 10 cells from 3 mice) and not different from the value in unstimulated mice (*p* = 0.36, Figure 4B) (Figure 4B). Such dramatic prolongation of τ EPSC suggests a longer presence of glutamate at the postsynaptic receptors consistent with the concept of compromised glial uptake.

### 2.3. Memantine Prevents the Loss EAAT1 after Chronic Optogenetic Stimulation of BG

The bulk of glutamate is taken up by EAAT1 located on BG processes [9]. After 4 days of photostimulation, EAAT1 immunoreactivity strongly decreased in BG (Figure 5A). The area of EAAT1 positive elements fell from 66.8 ± 5.3% (8 areas from 3 mice) to 21.3 ± 3.0% (9 areas from 3 mice) (*p* = 0.001005). Memantine restored the area of EAAT1 positive elements to 56.1 ± 5.4 (6 areas from 3 mice) (Figure 5B), a value not different from the unstimulated mice (*p* = 0.29). Thus, memantine largely prevents changes in EAAT1 expression after chronic photo-activation of BG expressing. Specifically, we found a wider presentation of glutamate uptake sites on the processes of BG in memantine-treated mice.

## 3. Discussion

The aim of this work was to investigate whether memantine could be protective in a model of cerebellar neurodegeneration induced by chronic optogenetic activation of ChR2 expressed in BG ([5] and Figure 1A–C). Initially, the impact is selectively directed against BG, and this model allows us to study how the pathologic astrocyte function affects the morphology and viability of PCs. Under physiological conditions, astrocytes are essential for the vital functions of neurons. In contrast, reactive glia can exacerbate or even evoke neurodegeneration [10,11]. 

ChR2 expression in BG in our model is achieved by using AVV with an enhanced GFAP promoter targeted to glial cells [12,13] (Figure 1D). Chronic activation of BG^ChR2+^ leads to qualitative changes in the cytoskeleton, manifested by an increased GFAP expression in astrocytes. Such changes clearly affect PC, resulting in a decreased number of calbindin-positive PCs (Figure 2A,B). Reactive BG^ChR2+^ induced by 4 days of photostimulation have thicker and shorter processes compared to normal BG, and this was significantly prevented by memantine. In memantine-treated mice, the thickness and length of BG processes were not statistically different from the control group (Figure 3E,F). The same pattern was registered for the PCs. Chronic photostimulation of BG significantly reduced the number of PCs and the length of their dendrites (thickness of ML). Consistently, the size of a PC membrane was reduced, as evidenced by a decrease in the capacitance of PCs (Table 1). In animals that received memantine, these parameters statistically did not differ from the control group (Figure 3G,H and Table 1).

After 4 days of photostimulation amplitude of PF EPSC, evoked by a single electric stimulus was significantly decreased, but other basic properties of synaptic transmission between PF and PC were relatively preserved (Table 1). We detected a dramatic prolongation of the τ of EPSC evoked by burst stimulation, and this suggests a longer presence of glutamate at the postsynaptic receptors. Using a burst, in this case, was motivated by the intention to load the glutamate uptake system by a bulk of glutamate released over this short period. It is evident that the uptake system was severely compromised by the chronic optogenetic activation of BG. Memantine administration partly prevented τ PF EPSC prolongation while it did not affect its amplitude (Figure 4A,B). Most likely, this reflects a better-preserved function of EAAT1.

We hypothesize that morphological changes in BG and PCs and alteration of transmission in PF-PC synapses are due to excitotoxicity after chronic Na+ overloading of BG through ChR2. This idea is supported by our previous data [5]. Transporters of excitatory amino acids, such as EAAT1 in BG, use Na^+^ gradient for glutamate transport [14]. An increase in intracellular Na^+^ due to repetitive opening of ChR2 is likely to overload cytosol with Na^+^ and slow down this process. Thus, glia fails to take up glutamate from the intracellular space efficiently. 

We quantified the expression of EAAT1 after chronic photostimulation using immunohistochemistry and observed reorganization of EAAT1 after chronic photostimulation. Clusters of EAAT1-positive elements were fewer. Immunofluorescence is not quantitative enough to judge the amount of EAAT1 protein under these conditions, but if the transporters in the stressed BG concentrate in some clusters instead of being localized in the immediate vicinity of the sites of glutamate release, this could be an interesting and novel mechanism limiting the efficacy of glutamate uptake, which we believe, is the explanation for the much longer decay of glutamatergic EPSC (Figure 4B).

How could memantine be preventive in our model? At this moment, it is not clear whether memantine may have a direct effect on BG. Evidence for the presence of functional NMDA receptors on BG is faint. Apparently, BG may express low amounts of NR1, NR2A, NR2C and NR3B subunits [15,16]. Whether these form functional receptors NMDA is unclear; at least a study by [17] reported that in neonatal mouse slices, currents evoked by NMDA were insensitive to Mg^2+^, were not enhanced by glycine, and NMDA did not evoke Ca^2+^ elevations in BG. These currents were, however, sensitive to ketamine. 

A study on the cortical astrocytes of mice [18] reported that in these cells, NMDA can evoke currents and that the NMDA receptors on these astrocytes have low Ca^2+^ permeability. In fact, memantine in that study was more effective against NMDA-mediated currents in astrocytes than in neurons (IC50 of 2.19 vs. 10.9 µM). 

Therefore, at present, there is no convincing evidence for functional NMDA receptors on BG of adult mice, although this needs to be investigated directly.

The most probable scenario, therefore, is that memantine acts on PC, reducing their reaction to the excessive extracellular glutamate during chronic BG photostimulation. Neurodegeneration is clearly a bidirectional process, and the apoptotic events which obviously take place in PCs (as evidenced by their marked loss) may induce secondary reactivity in BG. Previously it has been shown that loss of PC in a transgenic model where mutant Ataxin 1 is selectively expressed in PC (SCA1 B05 model mice) can cause a reactive decrease in GLAST expression in BG [19]. It may be expected that a reduction of GFAP overexpression by memantine (Figure 2) also results from reduced pathologic signaling from the affected PC to the neighboring BG. Moreover, the microglia activation, which was not studied here, most likely contributes to the pathological outcome in this model. Finally, we cannot exclude that some effects of memantine are not related to the NMDA receptor function.

In summary, our data for the first time demonstrate the mechanism of potential therapeutic effect of memantine in a general model of the neurodegeneration in the cerebellum, which has features similar to spinocerebellar ataxia type 1 [5,20]. Oral administration of a clinically approved drug such as memantine to new forms of neurodegeneration could be quickly translated to the clinic. Memantine is usually well-tolerated, although it can cause some side effects. [21]. 

Therefore, data justify further research into the possible use of memantine in spinocerebellar ataxias and similar neurodegenerative diseases.

## 4. Materials and Methods

All procedures were carried out according to the Krasnoyarsk State Medical University, Russian Public Standard (33215–2014), U.K. Animals (Scientific Procedures) Act, 1986 and are in line with Helsinki protocol for handling experimental animals (WMA Statement on animal use in biomedical research, 2016).

### 4.1. Adenoviral Vector (AVV)

To achieve a high level of ChR2 expression in BG, we used AVV with an enhanced GFAP promoter [12]. The construct (GFAP-ChR2-mKate) was described previously [13,22]. In this vector, red fluorescent protein mKate is used as a marker for transduced cells. AVV had proliferated in HEK 293 cultures until full cytopathic effect, after which viral particles were released by sonication. The debris was removed by centrifugation (3000× *g* 10 min). AVV was purified using ultracentrifugation (Optima X, Beckmann Coulter, Brea, CA, USA) in CsCl gradient, aliquoted and stored at −80 °C.

### 4.2. Modeling Neurodegeneration

Thirty twelve-week-old WT CD1 mice were used in these experiments. The detailed protocol was described previously [5]. Briefly, Mannitol 25% 30 μg/mg of body weight was injected intraperitoneally in order to increase transduction. After 15 min, 10 μL of AVV GFAP-ChR2-mKate (3.7 × 10^7^ U/mL) were slowly injected into the cortex of cerebellar vermis (lobule VI) of deeply anesthetized mice using a 10 μL Hamilton syringe (Figure 1A). The stereotaxic coordinates relative to bregma were: AP: −2.5 mm, ML: 0 mm, DV: 2 mm. The mice were used for further experiments 4 days after the injection. To develop astrogliosis and neurodegeneration, 60 s long trains of pulses (20/20 ms on/off) of blue light with 60 s breaks were continuously applied for 3 days following AVV transduction. A cranial window of 2.5 × 2.5 mm was left in the skull above the cerebellum while keeping meninges intact to deliver light. A light diode was fixed by cement above it (Figure 1B). The wire from the diode to the controller was suspended without tension and via a connector which allowed free movements in the cage. In the control group, there was no light application. Memantine (Sigma-Aldrich, St. Louis, MO, USA, cat. No M9292) was added to drinking water at 0.5 mg/mL. On average, mice consume daily ~5.8 ± 0.2 mL/30 g of mouse body weight (consistent with Bachmanov et al., 2002 [23]). The drinking bottle was replaced once a week, and the intake was confirmed. Thus, the calculated dose of memantine was approximately 90 mg/kg/day. Administration of memantine commenced on the first day of photostimulation. This dose of memantine was tolerated well by mice and was previously reported to be effective against neurodegeneration in the brainstem [24]. 

### 4.3. Patch Clamp Recordings

Cerebellar slices (250 μm thick) were prepared, and whole-cell recordings were conducted as described previously [25]. Briefly, mice were deeply anesthetized by intraperitoneal injection of chloral hydrate (400 mg/kg of body weight) and killed by decapitation. The brains were quickly dissected and placed for one minute in an ice-cold Ringer’s solution containing the following: 234 mM sucrose, 26 mM NaHCO3, 2.5 mM KCl, 1.25 mM NaH_2_PO_4_, 11 mM glucose, 10 mM MgSO_4_, and 0.5 mM CaCl_2_ 0.5; pH 7.4, continuously oxygenated with 95% O_2_ and 5% CO_2_. Parasagittal slices of cerebellar vermis were made using a micro slicer (Thermo Scientific; Microtome CU65). The slices were maintained in an extracellular solution containing the following: 125 mM NaCl, 2.5 mM KCl, 2 mM CaCl_2_, 1 mM MgCl_2_, 1.25 mM NaH_2_PO_4_, 26 mM NaHCO_3_, 10 mM d-glucose and 0.05–0.1 mM picrotoxin (GABAA receptor blocker) bubbled by 95% O_2_ and 5% CO_2_ gas mix at room temperature for 1 h before starting the electrophysiological experiments. For voltage clamp whole-cell recordings from Purkinje cells (PCs) intracellular solution contained: 140 mM Cs-gluconate, 8 mM KCl, 10 mM HEPES, 1 mM MgCl_2_, 2 mM MgATP, 0.4 mM NaGTP and 0.4 mM EGTA (pH 7.3 adjusted with CsOH). Passive electrical properties of the PCs were estimated using hyperpolarizing voltage pulses (from −70 to −80 mV, 200 ms duration). The fast capacitance component was automatically compensated; the signal was sampled at 50 kHz and low-pass filtered at 10 kHz. No correction was made for liquid junction potentials. Analysis of electrophysiological data was performed using pClamp10 (Molecular Devices, San Jose, CA, USA), Patch Master software (HEKA, Ludwigshafen, Germany) and Clampfit 10.5 (Axon Instruments, Union City, CA, USA). PCs were voltage-clamped at −70 mV to record excitatory postsynaptic currents after activation of parallel fibers (PF EPSCs). Selective stimulation of PFs was confirmed by paired-pulse facilitation of EPSC amplitudes (at a 50 ms interstimulus interval). In experiments on slices containing ChR2-expressing BG, before patch clamp recordings, we first confirmed the presence of mKate fluorescence in the area. To assess the spillover effect of glutamate and its uptake by BG, we applied ten train stimuli to PFs with subsequent PF EPSC recording.

### 4.4. Immunohistochemistry

For immunohistochemistry (IHC), terminally anesthetized mice were perfused transcardially with 4% paraformaldehyde in 0.1 M phosphate buffer, brains were removed and postfixed in the same fixative overnight. The cerebellar vermis was cut into with 50 μm sagittal sections. The sections were treated with rabbit monoclonal anti-Calbindin D-28 k (1:500, Cloud Clone Corp., Wuhan, China), chicken polyclonal anti-GFAP antibodies (1:1000, Abcam, Cambridge, UK), rabbit polyclonal anti-EAAT1 (1:500, Abcam, Cambridge, UK), Secondary antibodies were Alexa Fluor 647-conjugated donkey anti-chicken IgG (1:1000, Life Technologies, Carlsbad, CA, USA) and Alexa Fluor 488-conjugated donkey anti-rabbit IgG (1:1000, Life Technologies, Carlsbad, CA, USA). Antibodies were dissolved in PBS solution containing 2% (*v*/*v*) donkey serum, 0.1% (*v*/*v*) Triton X-100 and 0.05% NaN_3_. 

### 4.5. Confocal Microscopy and Morphometric Analysis

In all groups, the cerebellar lobes 6 and 7 of the vermis cerebellum were used for comparisons. Fluorescent images were obtained using FV10i Confocal Microscope (Olympus, Tokyo, Japan). Images were recorded as Z-stacks using 10× objective. Images from the same confocal plane were compared to assess double labeling. The thickness and length of BG processes were measured on confocal images of sagittal cerebellar slices. The thickness of BG processes was analyzed using intensity profiles of a 100 μm line drawn across the layer where each glial process appeared as a peak of GFAP/Alexa 647 fluorescence, as described previously [5]. The number of PCs was measured across the PCL over the 100 μm stretch of the section. The approximate length of the dendrites of the Purkinje cells was estimated from the overall thickness of the molecular layer (ML), visualized using anti-Calbindin/Alexa 488 staining. 

EAAT1 expression was accessed using anti-EAAT1 fluorescence staining with Fiji software. The intensity of fluorescence was consistently lower in the sections from animals subjected to photostimulation and not treated by memantine. However, quantification of intensity across separate samples on confocal images is unreliable, and we decided not to do that. Instead, we converted the stained areas of the sections to binary images (8-bit greyscale, threshold 10 units with dark background, followed by “morphology” filter, to remove digital “dust”). By keeping the threshold the same across all groups (10 units) and converting all stained pixels to the same value (255), we avoided issues related to the brightness of the sections (10 units is essentially just above the level of digital noise). We then used the “Trace particles” function to measure the total number of individual stained objects, their average areas, their total area and related it to the overall area of the measured part of the cerebellum (Appendix A).

### 4.6. Statistical Analysis

Pooled data are expressed as M ± S.E.M. mean values with a 95% confidence interval. We used the R software to perform the statistical analyses. The differences between the individual groups were calculated with one-way ANOVA analysis, followed by the post-hoc Tukey’s Honestly Significant Difference (HSD) test for the *p*-values adjustment. Differences were considered significant at *p* < 0.05.

## 5. Conclusions

In a novel model of cerebellar neurodegeneration caused by chronic optogenetic activation of BG with ChR2, we have revealed a protective effect of memantine. This highlights the potential of memantine as a neuroprotector in cerebellar neurodegenerative diseases.

## Figures and Tables

**Figure 1 ijms-22-07822-f001:**
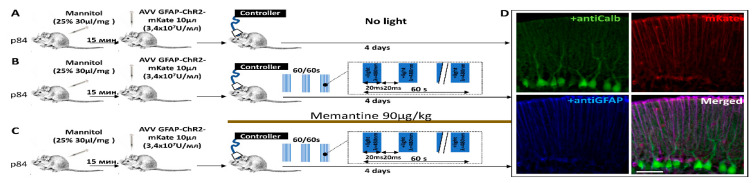
Optogenetic model of BG-induced neurodegeneration. Ten μL AVV GFAP-ChR2-mKate (3.4 × 10^7^ U/mL) was injected into the cerebellar cortex. A 20 mA light diode was cemented above the injection site. In the control group (**A**), no photostimulation was performed; in the experimental group (**B**), the cerebellar cortex was irradiated with pulsatile blue light (20/20 ms for 60 s) for 4 days. (**C**) After AVV injections, all animals received memantine, which was administered in the drinking water (~90 mg/kg daily). After 4 days, the cerebellum was removed and used for IHC. (**D**) Expression of AVV CFAP-ChR2-mKate was assessed by mKate fluorescence (red). This signal was co-localized with anti-GFAP (a marker of astrocytes) but not with an anti-Calbindin signal (a marker of PCs).

**Figure 2 ijms-22-07822-f002:**
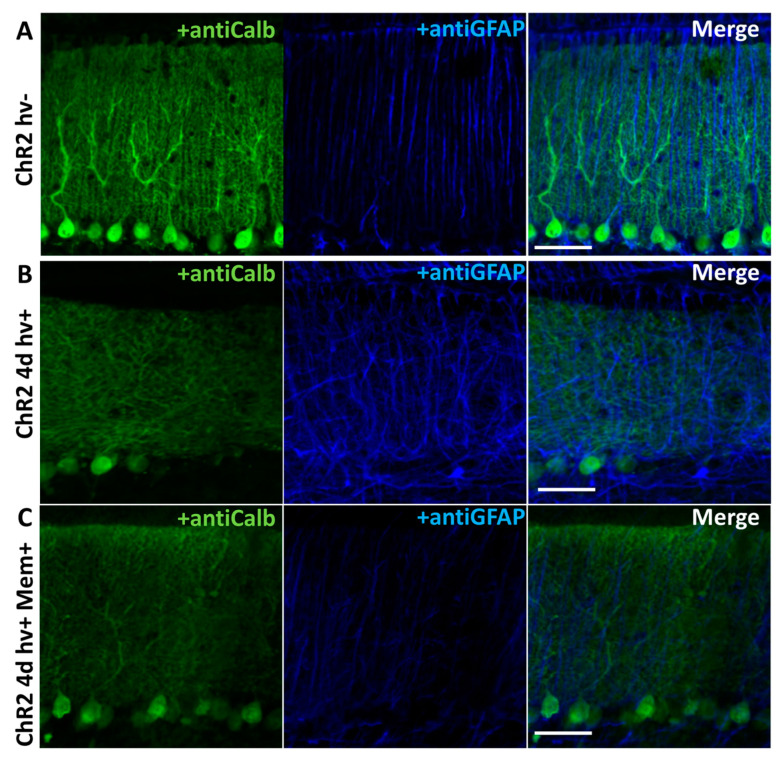
Memantine prevents changes in cerebellar cells morphology caused by chronic photostimulation of BGChR2. Confocal images of cerebellar cortex labeled with anti-Calbindin (PCs marker) and anti-GFAP (BG marker). (**A**) Animals without photostimulation (hv−). (**B**) Animals with chronic photostimulation, (hv+). (**C**) Animals with chronic photostimulation (hv+) treated with memantine (Mem+).

**Figure 3 ijms-22-07822-f003:**
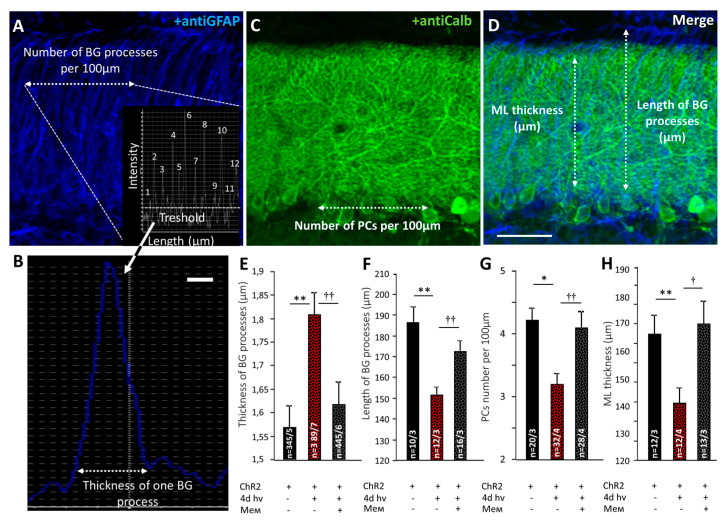
Memantine prevents the pathological transformation of BG and PCs caused by optogenetic hyperactivation of BG. Confocal images of cerebellar cortex labeled with anti-GFAP (glial marker) (**A**), anti-Calbindin (PCs marker) (**C**). The intensity profile of an anti-GFAP signal shows the number of peaks that mark the processes of BG (**A** insert). Such an approach leads to estimate the length of a single BG process (**B**). Scale bar 1 μm. The number of PCs was counted across the 100 μm line of PCL (**C**). BG processes were longer than the thickness of ML. These data were measures separately (**D**). Scale bar 50 μm. (**E**)—Averaged thickness of BG processes. (**F**) Averaged length of BG processes. (**G**)—Averaged number of PCs per 100 μm of PCL. (**H**)—Graph shows ML thickness 4 days after photostimulation. BG processes are thicker, shorter and twisted. Memantine prevents the pathological transformation of BG processes. * *p* < 0.05; ** *p* < 0.01; † *p* < 0.05; †† *p* < 0.01.

**Figure 4 ijms-22-07822-f004:**
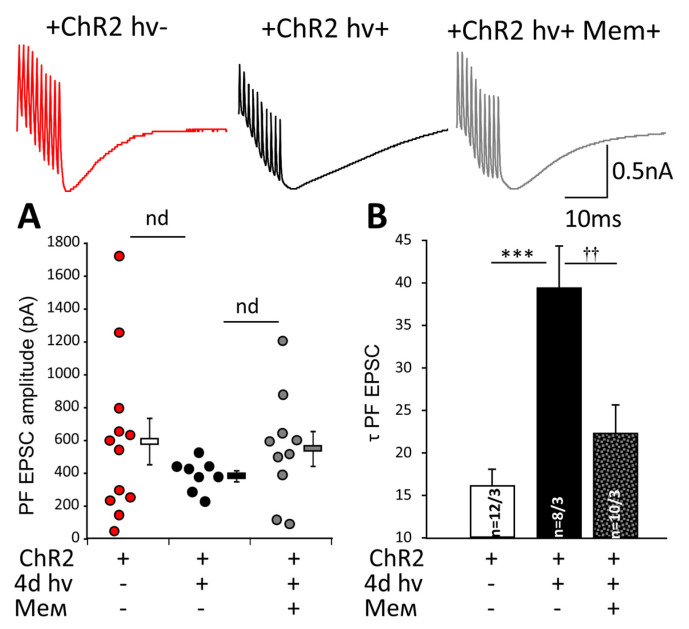
Chronic photostimulation of BGChR2 affects PF-PCs synaptic transmission. To assess the spillover effect of glutamate and its uptake by BG, we used bursts of 10 stimuli applied to PFs. The single amplitudes of the cumulative response to the burst of 10 stimuli were plotted to the graph, and averaged data were not significantly different in all examined groups (**A**). Prolongation of τ EPSC in PCs in animals after chronic photostimulation was prominent and significantly reduced in memantine-treated animals (**B**) *** *p* < 0.001; †† *p* < 0.01. Upper inserts: representative traces of PF EPSCs in animals expressing ChR2 without (ChR2+ hv−) (red) and after 4 days of photostimulation (ChR2+ hv+) (black) with oral memantine application (ChR2+ hv+ mem+) (grey).

**Figure 5 ijms-22-07822-f005:**
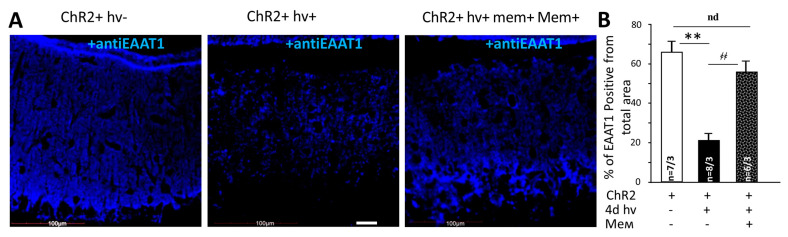
EAAT1 expression in reactive BGChR2+ and influence of memantine to this expression. (**A**) Confocal images of BG^ChR2+^ processes without (ChR2+ hv-) and after 4 days photostimulation (ChR2+ hv+) combined with oral administration of memantine (ChR2+ hv+ mem+). EAAT1 is labeled with blue color. Scale bar 20 μm. (**B**) The averaged surface of the EAAT1 positive elements in relation to the total area. ** *p* < 0.01; ҂҂ *p* < 0.01.

**Table 1 ijms-22-07822-t001:** Influence of memantine on basic passive electrophysiological and AMPA receptor-mediated EPSCs in PC in mice injected with AVV GFAP-ChR2-mKate after chronic photostimulation. Kinetics of PF EPSCs (10–90% rise time and decay time constant) and their amplitudes are shown. Paired-pulse facilitation (PPF) of PF EPSCs was analyzed using two stimuli with an interstimulus interval of 50 ms. Ratios of PPF were measured as the second EPSC amplitude normalized to the first one. One-way ANOVA followed by Tukey’s post hoc test indicates significant effects in Capacitance and Amplitude. *n*—number of cells/number of animals, * *p* < 0.05; ** *p* < 0.001; ^†^
*p* < 0.001.

Group	Capacity (pF)	Ra (MΩ)	Rm (MΩ)	Amplitude of PF EPSC (pA)	PPF	Rise Time (ms)	Decay Time (τ)
**+GFAP-ChR2-mKate (hv−)** **(*n* = 8/3)**	725.1 ± 39.5	12.4 ± 0.6	186.3 ± 28.2	298.1 ± 64.6	1.8 ± 0.1	2.2 ± 0.3	16.0 ± 2.1
**+GFAP-ChR2-mKate (+ 4 d hv) (*n* = 10/3)**	494.2 ± 63.2 **	13.8 ± 1.4	211.5 ± 34.5	141.0 ± 23.1 *	1.8 ± 0.1	2.4 ± 0.4	21.0 ± 3.0
**+GFAP-ChR2-mKate (+ 4 d hv) + Mem (*n* = 7/3)**	682.2 ± 24.5 ^†^	14.3 ± 0.9	173.4 ± 33.6	195.0 ± 33.7	1.8 ± 0.1	2.2 ± 0.2	18.8 ± 2.0

## Data Availability

The data presented in this study are available on request from the corresponding author.

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
