# Peer review of "Protective Effect of Memantine on Bergmann Glia and Purkinje Cells Morphology in Optogenetic Model of Neurodegeneration in Mice"

_ijms, 2021, doi:10.3390/ijms22157822_

Round 1

Reviewer 1 Report

The research article entitled “Protective effect of memantine on Bergmann glia and Purkinje cells morphology in optogenetic model of neurodegeneration in mice” by Shuvaev et al. discussed the effect of memantine on Glia and purkinje cells for possibility to be used in the treatment of neurodegenerative diseases. They observed that mimemtine reduces expression of glutamate transporter EAAT1.The article is interesting and point out the potential use of mimentine in reducing loss of glutamate transporter and PC. The article is very interesting. However, before it could be considered for publication, authors need to incorporate suggestions of the reviewer and revise their manuscript accordingly.

General comments Sentence formation needs crosscheck. The statements should be written in continuity and as a single para. Grammatical mistakes need to be minimized.

Abstract section does not give proper information. It is better to polish the abstract so highlights all the information and topics covered in the manuscript.

Introduction

  • Ca2+ needs to be corrected and should be written as Ca2+. The same problem persists throughout the manuscript
  • In the abstract, authors mention glial glutamate transporter EAAT1. Its full form should need to be added in the abstract.
  • A statement, “potentially it could be suitable for treatment of SCA. This, however requires additional evidence, specifically focusing on the cerebellar neurodegeneration” in the introduction need to be rephrased so as to bring clarity of the thoughts.
  • At other place, “ere we tested whether memantin can reduce neurodegenerative effects of chronic optogenetic activation of BG..” Th sentence needs to be written as, “ The present study was undertaken ……………”It will act as a suitable background information for the paper.

Introductory para need to be revised for the information to avoid any confusion between the statements.

Section methodology:

Authors have clearly referenced to methodology, if adopted from other studies.

Section results:

The results are written in a good way. However, small paragraphs need to be merged and information should be presented in a precise manner and as continuity with proper reference to corresponding figures.

Section conclusion:

The section needs to be a bit more elaborative and in reference to results obtained and with proper reference to previous studies. It should also highlight importance of the study and future directions with possible limitations.

Author Response

We are very pleased to get the excellent comments about our work, thank you for your positive assessment.

We have made changes according to your comments as listed here:

  1. Abstract part was reorganized in more logical way. We put the general information about spinocerebellar ataxias.
  2. We corrected all mistakes in introduction and in the text.
  3. We revised the introduction to avoid any confusion between the statements.
  4. We revised the conclusion and added the section to highlight the importance of this study.
  5. We merged small paragraphs and list the Fig. numbers in the proper way.

6. In section conclusions we included consideration of advantages and disadvantages of memantine treatment in clinical settings.

Reviewer 2 Report

The authors investigate the effect of memantine on Bergmann glia and Purkinje cells morphology in a recently developed optogenetic mice model of spinocerebellar ataxia (SCA). The authors have used a previously developed optogenetic approach to induce reactive phenotype associated with reduced EAAT1 activity in cerebellar Bergmann glia (BG) cells, resulting in increased excitotoxicity and death in Purkinje cells (PCs). The main result of the work is that oral administration of memantine (an uncompetitive antagonist of NMDA receptors) partially prevented the reactive morphological phenotype of BG induced by the photostimulation in the mice model of SCA. Besides, preserving the BG phenotype was associated with EAAT1 normalization on BG and reduced PCs' loss.
The article presents exciting results as well as a new mice model for SCA research. It is appropriately presented and structured, including an excellent scientific discussion and comparison with the previously published data. However, I have some comments that authors should address:
1-Please corrects Bergman glia by Bergmann glia in the abstract and the introduction.
2-Can we exclude a direct morphologic or even molecular effect of chronic photostimulation on PCs?
3-Can the authors include a histologic image (hematoxylin and eosin) for each panel (A. B, and C) of Figure 2. It would be interesting to compare the decrease and recovery of PCs after chronic photostimulation and memantine administration, respectively.
4-Please correct: "untreated (hv+)" in the Figure 2 legend.
5-Discussion, 6th paragraph, line 199-200: "At this moment it is not clear effect whether memantine may have a direct on BG." Change by "At this moment, it is not clear whether memantine may have a direct effect on BG."
6-Since excitotoxicity is mediated by oxidative deregulation in many neurodegenerative models, did the authors measure any cellular biomarker of oxidative stress/damage in any of the experiments carried out? Excitotoxicity leading to Ca2+ transient deregulation can activate some metabolic pathways resulting in free radical overproduction and neuronal cell death.
7-Discussion: It is unclear if the investigation addressed the action of memantine on any specific glial glutamate transporter subunit. Besides, can the authors elaborate a little bit on the seventh and eighth paragraphs (Lines 205-213)?

Author Response

Thank you very much for positive comments. We are very pleased that our work could be of interest to other researchers.

We carefully revised the manuscript and our remarks put here:

  1. Corrected "Bergman" to "Bergmann" glia in abstract.
  2. We used chronic photostimulation of ChR2 in astrocytes to develop excitotoxicity and cerebellar neurodegeneration. The precise methodology and negative effects were described in our previous work (Shuvaev et al., 2021). We used PBS injected mice with subsequent chronic photostimulation to exclude negative light effects that are not related to ChR2.
  3. Thank you very much for this comment. We are also thinking that PCs morphology can clearly show the positive effects of Memantine. However, Hematoxillin-eosine is not compatible with our immunofluorescent protocols and we cannot now perform additional series specifically for that purpose. However, the immunofluorescent confocal images of PC using calbindin staining are very detailed and allow very accurate assessment of the structure of these cells (Fig. 2). In addition we use patch clamp to electrically isolate and estimate the size of the PCs (See table 1).
  4. We corrected the legend of figure 2.
  5. Thank you very much for the comment. We changed this part.
  6. We thank the reviewer for this suggestion. Oxidative stress is linked with excitotoxicity and we will look into that matter in our next experiments.

7. We added the information about NMDA receptors subunits and their interaction with Memantine in discussion.

Round 2

Reviewer 1 Report

Authors have successfully answers all my queries. The MS can be accepted for publication.